# Coronary Artery Disease in Patients Hospitalized for Peripheral Artery Disease: A Nationwide Analysis of 1.8 Million Patients

**DOI:** 10.3390/diagnostics13061163

**Published:** 2023-03-18

**Authors:** Magnus Helmer, Christian Scheurig-Muenkler, Verena Brandt, Christian Tesche, Stefanie Bette, Florian Schwarz, Thomas Kroencke, Josua A. Decker

**Affiliations:** 1Department of Diagnostic and Interventional Radiology, University Hospital Augsburg, 86156 Augsburg, Germany; 2Department of Cardiology, German Heart Centre Munich, 80636 Munich, Germany; 3Department of Cardiology, Munich University Clinic, Ludwig-Maximilians-University, 80377 Munich, Germany; 4Department of Cardiology, Augustinum Clinic Munich, 81375 Munich, Germany; 5Medical Faculty, Ludwig-Maximilians-University Munich, 80336 Munich, Germany; 6Centre for Advanced Analytics and Predictive Sciences (CAAPS), University of Augsburg, 86159 Augsburg, Germany

**Keywords:** peripheral artery disease, coronary artery disease, in-hospital treatment, nationwide analysis, comorbidity

## Abstract

Purpose: Coronary artery disease (CAD) and peripheral artery disease (PAD) are highly prevalent in society. This nationwide analysis aimed to evaluate the trends of in-hospital treatment of patients admitted due to PAD with and without concomitant CAD, to determine the prevalence and risk factors of concomitant CAD in patients with PAD. Methods: Using data from the German Federal Statistical Office, we included all admissions for PAD (with and without concomitant CAD) in Germany between 2009 and 2018. Baseline patient characteristics, outcomes and comorbidities were analyzed. Elixhauser comorbidity groups and the linear van Walraven comorbidity score (vWs) were calculated to assess the comorbidity burden. Results: Of all 1,793,517 patients hospitalized for PAD, a total of 21.8% (390,259) had concomitant CAD, increasing from 18.6% in 2009 to 24.4% in 2018. Patients with accompanying CAD showed higher in-hospital mortality (3.7 vs. 2.6%), more major amputations (9.0 vs. 7.7%) and more comorbidities (Elixhauser score: 4.2 vs. 3.2 and vWs: 9.1 vs. 6.1), resulting in higher costs (median: EUR 4541 vs. EUR 4268 per case). More advanced stages of PAD were associated with multi-vessel CAD (10% of all patients with PAD Fontaine IV showed 3-vessel CAD) and the prevalence of multi-vessel CAD increased predominantly in patients with advanced PAD. Conclusion: One in four patients hospitalized for PAD had concomitant CAD, showing an increase over time with an additional medical and economic burden for hospitals compared with patients without CAD.

## 1. Introduction

Atherothrombotic diseases and their most common forms, coronary artery disease (CAD) and peripheral artery disease (PAD), are highly prevalent in society [1,2]. They are the main cause of death and morbidity worldwide, imposing significant medical and economic costs [3,4]. Over 200 million people are estimated to be affected by PAD, with a global prevalence of up to 5% [5,6,7]. PAD patients suffer much higher morbidity and mortality than unaffected people, with estimates of a 2- to 6-fold increase in the risk of cardiovascular events [8,9,10]. Given the aging of our population due to medical advances, an upward trend of these conditions is to be expected [11].

Both CAD and PAD are based on a common pathophysiology [1,2] and share, besides age, the same lifestyle-associated risk factors for development: dyslipidemia, arterial hypertension and diabetes mellitus [12]. A recent paper took a closer look at patients suffering from PAD and CAD separately, according to the latest data available [13]. It illustrated the high medical burden and direct expenses of atherothrombotic disease [13], even though significant progress has been made in improving survival, morbidity and quality of life by the progressive usage of antithrombotic medication and drug/lifestyle adjustments to reduce modifiable cardiovascular risk factors [14] with also a recent increase in endovascular interventions. However, large-scale data comparing hospitalizations due to PAD with and without CAD are, to our knowledge, lacking. 

Therefore, we sought to analyze commonalities and key differences in terms of risk factors, comorbidities, hospitalization time, procedures needed during hospitalization and cost between these two patient groups.

## 2. Materials and Methods

The detailed procedure for data acquisition was reported previously [15]. In brief, data of all hospitalizations due to PAD from 2009 to 2018 were acquired from the diagnosis-related group (DRG) statistics provided by the German Federal Statistical Office’s Research Data Center (RDC) [16]. The authors used data structure files to write syntaxes in R version 4.1.2 (R Foundation for Statistical Computing, Vienna, Austria, https://www.R-project.org/, accessed on 1 December 2021). Fully anonymized data were then submitted back to the authors.

### 2.1. Patient Cohort

From 2009 to 2018, we included all hospitalizations with PAD of the lower extremity as the primary diagnosis, using the German modification of the International Classification of Diseases (10th revision) (ICD-10-GM) codes of atherosclerosis of the arteries of the extremities, as follows: I70.21 (2009–2014) and I70.22 (2015–2018), pain-free walking distance of 200 meters or less (Fontaine IIb, Rutherford 2–3); I70.22 (2009–2014) and I70.23 (2015–2018), pain at rest (Fontaine III, Rutherford 4); I70.23 (2009–2014) and I70.24 (2015–2018), minor tissue loss with ulcers (Fontaine IV, Rutherford 5); and I70.24 (2009–2014) and I70.25 (2015–2018), major tissue loss with gangrene (Fontaine IV, Rutherford 6). Individually performed procedures such as amputations were identified by their respective codes of the Operation and Procedure Classification System (OPS).

Within all hospitalizations due to PAD between 2009 and 2018, we identified patients with concomitant CAD, represented by ICD-10 codes as follows. I25.10: atherosclerotic heart disease, without hemodynamically significant stenosis; I25.11: atherosclerotic heart disease, single-vessel coronary artery disease; I25.12: atherosclerotic heart disease, double-vessel coronary artery disease; I25.13: atherosclerotic heart disease, triple-vessel coronary artery disease; I25.14: stenosis of the left main coronary artery; I25.15: CAD with stenosis of a bypass vessel; and I25.16: CAD with in-stent stenosis. Furthermore, if CAD was present, we identified coronary interventions for CAD that were performed during hospitalization using the following OPS codes: 8-837 for percutaneous coronary interventions (PCIs) in general; 8-837.00 and 8-837.01 for coronary balloon angioplasty; 8-837.k for coronary stenting without medication (bare-metal stent); and 8-837.m for coronary stenting with medication (drug-eluting stent). For further analyses, the largest subgroups of non-stenosing CAD (CAD 0) and 1- to 3-vessel disease (CAD 1–CAD 3) were separately analyzed.

### 2.2. Elixhauser Comorbidity Groups and van Walraven Score 

Secondary diagnoses were separately assessed and categorized into Elixhauser comorbidity groups, according to Elixhauser et al. [17]. To assess and evaluate the extent of comorbidities and in-hospital mortality, the Elixhauser score (sum of positive Elixhauser groups) and weighted linear van Walraven comorbidity score (vWs) were calculated accordingly using the established ICD-10 coding definition of Quan et al. [18,19].

### 2.3. Statistical Analysis

Data were analyzed using R version 4.1.2 (R Foundation for Statistical Computing, Vienna, Austria, https://www.R-poject.org/, accessed on 1 December 2021) and were provided as numbers with percentages, the mean and standard deviation (SD) for normally distributed data or the median with the interquartile range (IQR) for non-normally distributed data. The Elixhauser comorbidity groups, Elixhauser score and weighted linear van Walraven score were calculated using the R package of comorbidity (comorbidity: Computing Comorbidity Scores. Available online: https://cran.r-project.org/package=comorbidity, accessed on 1 December 2021) [20]. Due to the large size of the dataset, very small (and potentially negligible) differences between groups yielded highly significant *p*-values. Therefore, we refrained from statistical tests and the reporting of p-values and opted for a descriptive presentation of our data.

## 3. Results

### 3.1. Baseline Characteristics

A total of 1,793,517 (1,133,725 (63.2%) male) hospitalizations due to PAD (Fontaine stage IIb and higher) were included in this study. Concomitant CAD was observed in 390,259 hospitalizations (21.8%). Between 2009 and 2018, we observed a 48.8% increase in hospitalizations with any concomitant CAD (from 18.6% (30,356) to 24.4% (45,160)). Non-stenosing CAD decreased from 1.5% to 1.2%, whilst 1- to 3-vessel disease steadily increased (Figure 1). 

Men accounted for 60.9% of patients without and 71.7% of patients with CAD, a difference of 17.7%. The mean age was slightly higher in patients hospitalized with concomitant CAD compared with hospitalizations without concomitant CAD (72.7 vs. 71.0 years). The rate of in-hospital mortality was higher in hospitalizations with concomitant CAD compared with those without CAD (3.7% (14,524) vs. 2.6% (35,955)). The rate of in-hospital mortality decreased in both patients with only PAD and patients with concomitant CAD, by 20.8% (from 3820 in 2009 to 3185 in 2018) in patients with PAD and by 10.2% (from 1200 in 2009 to 1603 in 2018), respectively. Thus, the ratio in-hospital mortality of patients with concomitant CAD compared with patients without CAD increased by 13.4% (1.38 to 1.54) between 2009 and 2018. Invasive ventilation was performed more often in hospitalizations with CAD compared with hospitalizations without PAD (2.7 (10,678) vs. 1.4% (20,317)).

The median reimbursement costs of hospitalizations with CAD were higher than those without CAD (EUR 4541 (2669–7735) vs. EUR 4268 (2419–6964)). Although reimbursement costs for CAD and non-CAD showed a similar development between 2009 and 2013, we observed a pronounced gap starting in 2014, with increasing reimbursement costs of hospitalizations with PAD and concomitant CAD whereas the costs of non-CAD patients remained constant (Figure 2). Detailed data are presented in Table 1.

### 3.2. Stages of PAD and CAD

The prevalence of more severe PAD stages increased with the severity of CAD; hence, in patients with PAD and no significant stenosis, CAD had the lowest prevalence and accounted for less than 2% (24,242). The co-prevalence increased, however; up to 9% (161,658) in 3-vessel coronary artery disease in the same patient group. Between 2009 and 2018, we documented a relative increase in all stages of CAD, except for the mildest form (no significant stenosis) with a decrease of 20.8%. The most striking increase was observed in patients with 3-vessel CAD, from over 7% (11,727) in 2009 to more than 10% (19,583) in 2018 (Figure 1). Detailed data of the PAD/CAD co-prevalence are provided in Table 2 and data on annual co-prevalence are presented in Appendix A. 

### 3.3. Comorbidities

Hospitalizations with accompanying CAD showed a higher rate of comorbidities, reflected by a higher Elixhauser score (4.2 ± 1.8 vs. 3.2 ± 1.6) and higher van Walraven score (9.1 ± 7.3 vs. 6.1 ± 6.1). Detailed data on the different comorbidities according to the Elixhauser groups are provided in Table 3. Patients with CAD showed a higher proportion of various comorbidities, with valvular diseases (factor 2.9) and congestive heart failure (factor 2.5) being the most prominent.

### 3.4. Individual Procedures

We observed minor differences between different revascularization approaches for PAD between the groups with and without concomitant CAD (Table 1). Between 2009 and 2018, we observed increasing endovascular revascularization approaches in both groups (concomitant CAD and non-concomitant CAD) whereas surgical revascularizations decreased. Furthermore, more minor (9.0 vs. 7.7%) and major amputations (4.3 vs. 4.2%) were performed in patients with both PAD and CAD compared with patients without concomitant CAD. Between 2009 and 2018, the rate of minor amputations slightly increased in patients with and without CAD (with CAD, from 2403 (7.9%) to 4163 (9.2%); without CAD, from 9780 (7.3%) to 10,949 (7.8%)). In contrast, the rate of major amputations decreased in patients with CAD from 5.4% (1625) to 3.7% (1653) and in patients without CAD from 5.6% (7406) to 3.4% (4770) (Figure 3).

PCIs were constantly performed on a small share (3.5%) of hospitalizations with accompanying CAD. Of these, coronary balloon angioplasty (1.6%) and the placement of stents were the most common procedures. Between 2009 and 2018, we observed a decrease in the use of bare-metal stents and a corresponding increase in drug-eluting stents (Figure 4).

## 4. Discussion

Including almost 1.8 million hospitalizations due to PAD in Germany between 2009 and 2018, this study investigated the differences in the hospitalizations of patients with and without concomitant CAD. The most important results were as follows: (1) the co-prevalence of CAD in patients hospitalized due to PAD was high and steadily increased, from below 20% in 2009 to almost 25% in 2018; (2) the in-hospital mortality in patients with additional CAD was almost twice as high compared with patients without CAD; (3) hospitalizations due to PAD with concomitant CAD resulted in overall higher treatment costs; and (4) patients with accompanying CAD showed overall higher comorbidities, reflected in increased Elixhauser and van Walraven comorbidity measures.

Therefore, and especially in light of an increasing prevalence of cardiovascular morbidities in our aging society, our data provides a thorough overview of the current state of in-hospital care and the differences in the treatment of PAD with and without concomitant CAD. As PAD and CAD are such common conditions in Germany, with a significant proportion of the population affected, effective stationary care can help to improve the quality of life and reduce the burden of these diseases on patients and the healthcare system [21,22]. Furthermore, PAD and CAD often occurring together highlights the importance of a comprehensive and integrated approach to the management of these conditions in stationary care [23].

As pointed out later, PAD and CAD require similar, not equal, treatment approaches; thus, it is important to ensure that patients receive the most appropriate care based on their individual needs and conditions [13,14]. Effective stationary care for PAD and CAD can help to improve patient outcomes, reduce the risk of major adverse events and promote overall health and wellbeing [24].

Our observations were in line with previous studies. PAD and CAD are age-related diseases [25,26,27]; an increasing occurrence of PAD was not surprising and is likely to reach new heights in the future. PAD and CAD share common pathogeneses and risk factors with hypertension, hypercholesterolemia and type 2 diabetes [1,2]. Our study showed these risk factors to be more prevalent in patients suffering from both PAD and CAD than in patients without concomitant CAD. Thus, not only is a high medical, but also a socioeconomic, burden in severe stages of PAD to be expected, the likelihood of greater morbidity as well as additional medical and financial resources needed due to higher stages of CAD increases. 

We reported a negative impact of additional CAD on the course of patient hospitalization, especially in terms of mortality, ventilation rates and minor and major amputations of the lower extremities. This was unlikely due to concomitant CAD alone; however, we found concomitant CAD to be additionally associated with a higher burden of other comorbidities such as type 2 diabetes mellitus, renal failure or valvular diseases. The higher comorbidity burden of patients with both PAD and CAD, which was expressed by increased Elixhauser and van Walraven comorbidity scores, might, therefore, be largely responsible for the reported adverse in-hospital outcomes. Although a steep increase in the comorbidity of CAD with other diseases directly related to the heart (e.g., congestive heart failure or cardiac arrythmias) is to be expected and was evidenced in our study, we also observed increased comorbidities not directly linked to the heart. However, many of these, such as hypertension and kidney failure, could be explained by the overall increased atherosclerotic burden in these patients. 

The treatment of PAD in patients with and without CAD showed no relevant difference and the well-known trend away from open surgical towards endovascular interventions [14,15] was similar in both groups. This was likely due to the fact that minimal invasive endovascular procedures are not hindered by the higher degrees of morbidity in CAD patients; in fact, the trend away from invasive vascular surgeries should be especially appreciated by a comparatively more morbid patient group, thus reducing individual peri-interventional risks. 

The overall DRG-related reimbursement of treatment costs in patients with concomitant CAD was higher; however, the sole treatment of PAD hardly contributed to this because, as already mentioned, the treatment methods for PAD in both patient groups remained almost the same. Rather, the additional costs were likely a result of necessary additional treatments for other comorbidities (such as CAD) in this almost two year older cohort.

It could be concluded that if the current trajectory of rising CAD in patients hospitalized with PAD is to continue, it will inevitably result in an even bigger strain being placed on the healthcare system. In turn, a reduction in the risk factors associated with CAD and PAD as mentioned might have the potential to slow down, perhaps even stop, the upward trend of these pathologies. Previous studies have shown that lifestyle choices can mitigate the mentioned risk factors and the use of antithrombotic medication can reduce the risk of both PAD and CAD; thus, secondary preventive strategies, including medication to manage modifiable CV risk factors and antithrombotic medication to prevent blood clot formation, will improve prognoses [13,28]. Medical professionals should be aware that upon the admission of PAD patients, there is a high chance of additional CAD. Patient screening for CAD on admission could lead to a better management of concomitant CAD. 

The strength of our study lay in its large sample size of 1.8 million patient data, which provided a robust representation of the patient population. The German healthcare system is comparable with other industrialized nations. The comparison of patients suffering from PAD with and without concomitant CAD was a unique and focused approach, which allowed for a more in-depth understanding. The vast sample size and clear focus on this issue increased the validity and reliability of our findings and made our study a valuable contribution to the field. Furthermore, this study provided a valuable opportunity to compare and validate previous findings and to identify new and important strategies of in-hospital treatment. Overall, the large sample size and focused approach made this study a strength in the field of patient data analysis.

Apart from the obvious advantages of a large, continuous real-world dataset, this study had certain limitations: (1) We were only able to analyze inpatient and not outpatient treatments using the data given. Whilst outpatient treatment for PAD is available, and occasionally preferred in the healthcare systems of other countries, PAD-related interventions are primarily conducted in hospitals in Germany. Thus, the total financial and medical burden of PAD with and without concomitant CAD in an outpatient setting was not reflected in this study. (2) Our data showed individual hospitalizations; we could not identify individual patients. This introduced a bias due to the potential multiple occurrences of individuals who required several hospitalizations. (3) The reimbursement did not accurately reflect the expenses of an individual hospital stay, but rather the average costs associated with a certain DRG across Germany. The real costs might even be higher in concomitant diseases in multi-morbid patients. (4) We analyzed administrative data collected by each treating institution for payment purposes. Errors in coding techniques or economic reasons may have introduced an unavoidable bias.

## 5. Conclusions

Between 2009 and 2018, the total number of PAD-related hospitalizations increased with a concurrent overproportionate increase in the prevalence of concomitant CAD. Hospitalizations for PAD with concomitant CAD showed an increased age, more comorbidities, higher costs and less favorable in-hospital outcomes, with increased in-hospital mortality and increased amputation rates.

## Figures and Tables

**Figure 1 diagnostics-13-01163-f001:**
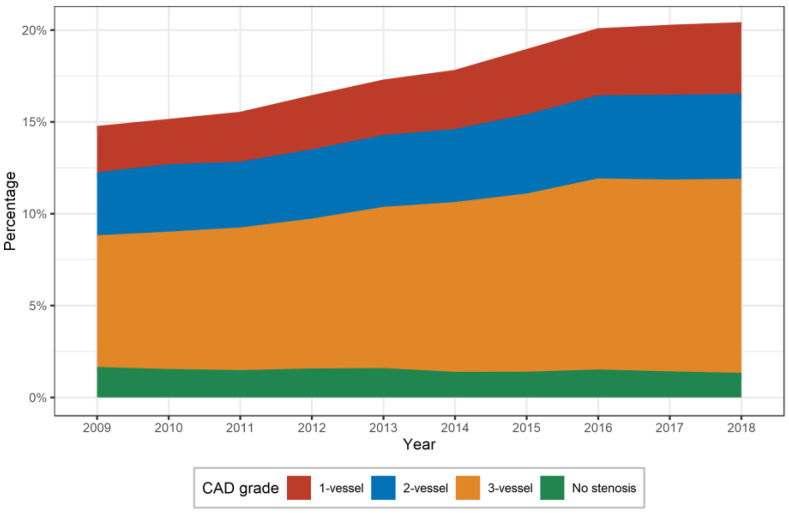
Prevalence of CAD (coronary artery disease) stages in patients hospitalized for peripheral artery disease in Germany between 2009 and 2018.

**Figure 2 diagnostics-13-01163-f002:**
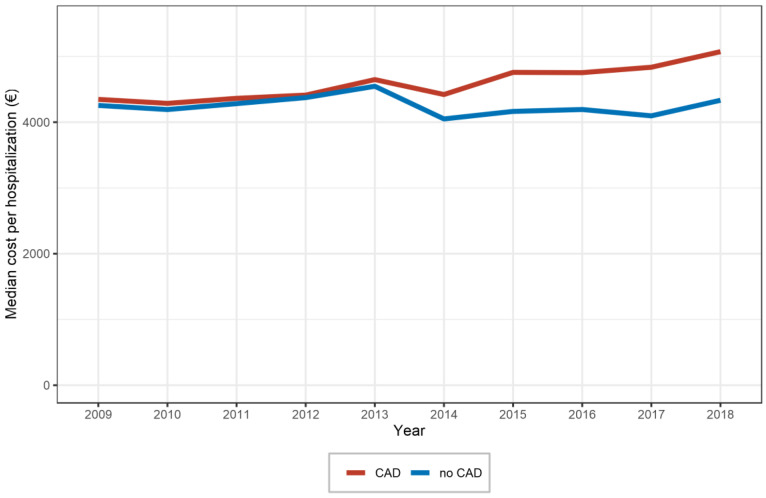
Median costs of hospitalizations for peripheral artery disease with and without concomitant coronary artery disease (CAD) in Germany between 2009 and 2018.

**Figure 3 diagnostics-13-01163-f003:**
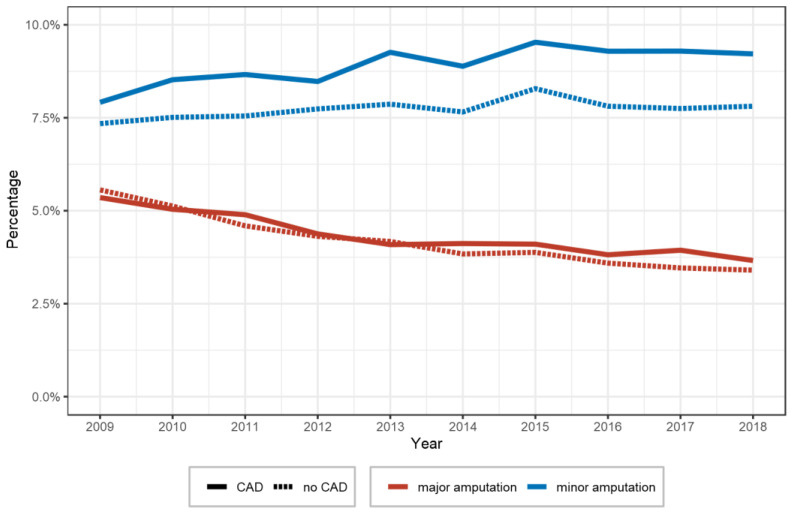
Prevalence of major and minor amputations in patients hospitalized due to peripheral artery disease with and without coronary artery disease between 2009 and 2018.

**Figure 4 diagnostics-13-01163-f004:**
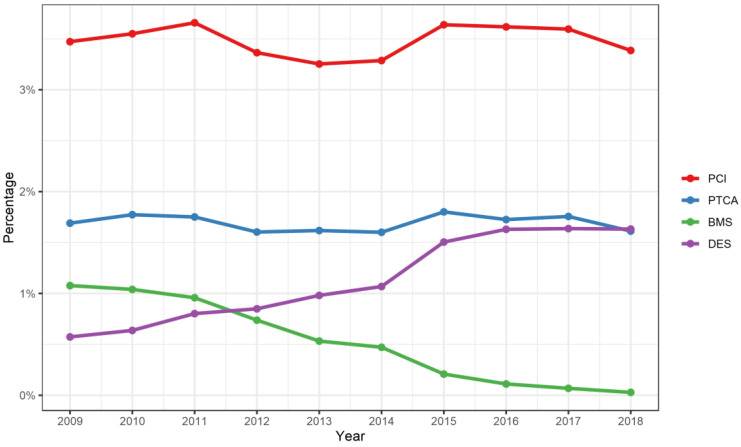
Prevalence of coronary interventions in patients hospitalized due to PAD between 2009 and 2018 in (%). PCI: percutaneous coronary intervention; PTCA: percutaneous transluminal coronary angioplasty; BMS: bare-metal stent; DES: drug-eluting stent.

**Table 1 diagnostics-13-01163-t001:** General characteristics of patients admitted for peripheral artery disease with and without coronary artery disease between 2009 and 2018.

Characteristic	CAD +	CAD −
Number	390,259 (21.8%)	1,403,258 (78.2%)
Age, years	72.7 ± 9.9	71 ± 11.5
Sex, male	279,759 (71.7%)	853,966 (60.9%)
In-hospital mortality	14,524 (3.7%)	35,955 (2.6%)
Major amputation	16,661 (4.3%)	58,707 (4.2%)
Minor amputation	34,965 (9.0%)	108,501 (7.7%)
Ventilation	10,678 (2.7%)	20,317 (1.4%)
In-hospital stay, days	7 (3–15)	7 (2–13)
Cost per hospitalization, EUR	4541 (2669–7735)	4268 (2419–6964)
Elixhauser score	4.2 ± 1.8	3.2 ± 1.6
van Walraven score	9.1 ± 7.3	6.1 ± 6.1
CVRF Hypertension Dyslipidemia Type II DM Smoking	280,307 (71.8%) 206,922 (53%) 176,178 (45.1%) 36,222 (9.3%)	896,795 (63.9%) 461,530 (32.9%) 435,735 (31.1%) 174,777 (12.5%)
PAD Revascularization None Endovascular Surgical Hybrid Two-step	407,144 (29.0%) 643,232 (45.8%) 246,992 (17.6%) 58,064 (4.1%) 47,826 (3.4%)	113,572 (29.1%) 172,712 (44.3%) 71,796 (18.4%) 17,150 (4.4%) 15,029 (3.9%)

Data are presented as number (percentage), mean ± standard deviation or median (interquartile range). PAD: peripheral artery disease; CAD: coronary artery disease; CVRF: cardiovascular risk factor; DM: diabetes mellitus.

**Table 2 diagnostics-13-01163-t002:** Prevalence of patients hospitalized for peripheral artery disease with and without concomitant coronary artery disease.

	Fontaine IIa	Fontaine III	Fontaine IVu	Fontaine IVg
**Male** No CAD CAD 0 CAD 1 CAD 2 CAD 3	461,467 (81.4%) [−5.5%] 8153 (1.4%) [−15.0%] 19,283 (3.4%) [34.3%] 24,597 (4.3%) [23.3%] 53,617 (9.5%) [36.8%]	107,434 (78.7%) [−7.9%] 2006 (1.5%) [3.8%] 4905 (3.6%) [47.0%] 6649 (4.9%) [35.0%] 15,568 (11.4%) [38.2%]	148,845 (78.8%) [−10.9%] 2617 (1.4%) [−39.4%] 6061 (3.2%) [97.9%] 8518 (4.5%) [62.2%] 22,852 (12.1%) [62.9%]	188,519 (78.2%) [−11.7%] 3057 (1.3%) [−35.0%] 7846 (3.3) [88.0%] 11,004 (4.6%) [55.5%] 30,727 (12.7%) [62.9%]
**Female** No CAD CAD 0 CAD 1 CAD 2 CAD 3	242,121 (87.5%) [−3.3%] 3304 (1.2%) [−16.4%] 7973 (2.9%) [52.5%] 9027 (3.3%) [23.9%] 14,133 (5.1%) [28.4%]	82,679 (85.4%) [−4.4%] 1267 (1.3%) [−7.0%] 3025 (3.1%) [59.8%] 3483 (3.6%) [23.9%] 6375 (6.6%) [35.8%]	124,263 (86.9%) [−5.9%] 1642 (1.1%) [−11.8%] 3932 (2.7%) [88.5%] 4352 (3.0%) [45.0%] 8883 (6.2%) [52.2%]	124,256 (86.8%) [−5.3%] 1535 (1.1%) [−32.1%] 3622 (2.5%) [64.5%] 4327 (3.0%) [50.9%] 9488 (6.6%) [53.1%]

Data are presented as absolute number, (percentage) and [relative change] between 2009 and 2018. PAD: peripheral artery disease; CAD: coronary artery disease; Fontaine IVu: Fontaine IV with ulcers; Fontaine IVg: Fontaine IV with gangrene; CAD 0–3: number of vessels with significant stenoses.

**Table 3 diagnostics-13-01163-t003:** Distribution of Elixhauser groups among patients hospitalized for peripheral artery disease with and without concomitant coronary artery disease between 2009 and 2018.

Elixhauser Group	CAD +	CAD −	Factor
Hypertension (uncomplicated)	277,535 (71.1%)	890,576 (63.5%)	1.1
Renal failure	154,943 (39.7%)	324,156 (23.1%)	1.7
Cardiac arrhythmias	126,924 (32.5%)	236,102 (16.8%)	1.9
Congestive heart failure	104,209 (26.7%)	152,640 (10.9%)	2.5
Diabetes (complicated)	98,014 (25.1%)	220,861 (15.7%)	1.6
Diabetes (uncomplicated)	75,787 (19.4%)	214,371 (15.3%)	1.3
Fluid and electrolyte disorders	61,526 (15.8%)	179,372 (12.8%)	1.2
Chronic pulmonary disease	58,709 (15.0%)	139,836 (10.0%)	1.5
Hypertension (complicated)	49,854 (12.8%)	75,571 (5.4%)	2.4
Valvular disease	41,376 (10.6%)	51,642 (3.7%)	2.9
Hypothyroidism	36,814 (9.4%)	112,075 (8.0%)	1.2
Obesity	35,640 (9.1%)	92,169 (6.6%)	1.4
Coagulopathy	20,394 (5.2%)	50,698 (3.6%)	1.4
Other neurological disorders	12,549 (3.2%)	43,941 (3.1%)	1.0
Depression	12,399 (3.2%)	39,558 (2.8%)	1.1
Paralysis	11,224 (2.9%)	42,696 (3.0%)	0.9
Pulmonary circulation disorders	9777 (2.5%)	14,235 (1.0%)	2.5
Deficiency anemia	8315 (2.1%)	21,467 (1.5%)	1.4
Rheumatoid arthritis/collagen vascular disorders	7068 (1.8%)	22,434 (1.6%)	1.1
Liver disease	5986 (1.5%)	18,068 (1.3%)	1.2
Alcohol abuse	5869 (1.5%)	29,292 (2.1%)	0.7
Weight loss	5775 (1.5%)	20,863 (1.5%)	1.0
Solid tumor without metastasis	4862 (1.2%)	17,244 (1.2%)	1.0
Blood loss anemia	1856 (0.5%)	5307 (0.4%)	1.3
Drug abuse	1021 (0.3%)	4270 (0.3%)	0.9
Metastatic cancer	959 (0.2%)	4193 (0.3%)	0.8
Psychoses	652 (0.2%)	3910 (0.3%)	0.6
Peptic ulcer disease, excluding bleeding	635 (0.2%)	1928 (0.1%)	1.2
Lymphoma	530 (0.1%)	1895 (0.1%)	1.0
AIDS/HIV	113 (0.0%)	382 (0.0%)	1.1

Data are presented as number (percentage). PAD: peripheral artery disease; CAD: coronary artery disease.

## Data Availability

The data underlying this article will be shared on reasonable request to the corresponding author.

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
