# Peer review of "Coronary Artery Disease in Patients Hospitalized for Peripheral Artery Disease: A Nationwide Analysis of 1.8 Million Patients"

_diagnostics, 2023, doi:10.3390/diagnostics13061163_

Round 1

Reviewer 1 Report

The article is very interesting and well-written. The statistical analysis is appropriate. The tables and figures are clear to understand. Congratulation to the authors!

Author Response

R1.1:
Comments and Suggestions for Authors

The article is very interesting and well-written. The statistical analysis is appropriate. The tables and figures are clear to understand. Congratulation to the authors!

Response to R1.1:
Thank you so much for taking the time to read and provide such positive feedback for our paper. Your kind words are greatly appreciated.

Reviewer 2 Report

To the authors,

   In this single-center, cross-sectional, retrospective, observational study, Magnus Helmer, et al showed patients with CAD had higher incidences of in-hospital mortality, major amputations and comorbidities, and resulted in higher costs. Moreover, advanced stages of PAD were associated with multi-vessel CAD. The prevalence of multi-vessel CAD was greater in patients with advanced PAD. These results were derived from the cohort with quite large number of patients, might be meaningful to estimate real-world daily clinical practice. However, several remaining concerns exist which need to be addressed appropriately.

1)    (Page 3, line 104) Although the authors were mentioned that the prevalence of hospitalizations with CAD increased from 18.6% to 24.4%, the result was not found in Figure 1. Please fix the figure as fitting to the explanation of the main text. 

2)    In the tables, the way of statistical analysis was insufficient. For example, the authors need to perform a student’s t test for normally distributed continuous variables and a Chi-square test for categorical variables. The current method for statistical analysis needs to be recheck and reassessed appropriately.

3)    (Page 5, line 138) Although the authors mentioned that “CAD had the lowest prevalence and accounted for less than 2 % (24,242); co-prevalence increased, however, up to 9 % (161,658) 139 in 3-vessel coronary artery disease in the same patient group.”. However, these results were not found in the figures or tables. The authors should show them in the figures or tables.

4)    The authors need to create a table or figure regarding patient in-hospital mortality and the details of the death, such as cardiac cause.

5)    What is the exact definition of CAD in this paper? It should be clearly mentioned in the method section. 

6)    (Page 8, line 200) I can’t understand “stationary care”. Please explain the term appropriately. 

7)    Please add sufficient explanation about the reason why the costs of hospitalizations with CAD were higher than those without CAD. Was it due to additional intervention procedure for CAD or additional treatment for their comorbidities? The details need to be added.

Since atherosclerosis is a systemic chronic inflammatory vascular disease, it is reasonable that severe PAD patients have greater incidence of CAD as well. Indeed, the conclusion derived from this study seems reasonable but have already shown in the similar studies in the past. What were new findings of the study compared with previous reports? This point needs to be clearly mentioned in the discussion section.

Author Response

Reviewer 2:

Comments and Suggestions for Authors

To the authors,

In this single-center, cross-sectional, retrospective, observational study, Magnus Helmer, et al showed patients with CAD had higher incidences of in-hospital mortality, major amputations and comorbidities, and resulted in higher costs. Moreover, advanced stages of PAD were associated with multi-vessel CAD. The prevalence of multi-vessel CAD was greater in patients with advanced PAD. These results were derived from the cohort with quite large number of patients, might be meaningful to estimate real-world daily clinical practice. However, several remaining concerns exist which need to be addressed appropriately.

R2.1:
(Page 3, line 104) Although the authors were mentioned that the prevalence of hospitalizations with CAD increased from 18.6% to 24.4%, the result was not found in Figure 1. Please fix the figure as fitting to the explanation of the main text

Response to R2.1:
We thank Reviewer 2 for this attentive observation. Our wording was indeed misleading as the term CAD in general includes more than just zero to three-vessel disease. There are also patients with stenoses in bypass vessels or stents. These small groups of other forms of CAD also need to be included in the greater picture (overall analysis). However, to show the prevalence and trends in the largest groups, we specifically investigated patients with zero to three-vessel disease. Therefore, we chose to add this figure only for these CAD grades to provide the reader with a clear graph.

For clarification, we specifically added this information in the methods section, and we clarified it in the results section before the figure.

R2.2:
In the tables, the way of statistical analysis was insufficient. For example, the authors need to perform a student’s t test for normally distributed continuous variables and a Chi-square test for categorical variables. The current method for statistical analysis needs to be recheck and reassessed appropriately.

Response to R2.2:
Thank you for your comment. Please let us explain our reason behind this approach:

We already performed several analyses using this kind of data:

https://link.springer.com/article/10.1007/s00330-021-08285-y
https://www.mdpi.com/2077-0383/11/7/2008
https://link.springer.com/article/10.1007/s00270-022-03136-9

Using such a large data set, we found any small difference to yield highly significant p-values. Any comparison between the two groups in Table 1 yields p<0.001. However, we think (and it has been stated by reviewers before) that this can also be misleading as insignificant differences (e.g. in-hospital stay of 7 vs 7 days in median) would also result in a p-value <0.001. We therefore chose to let the data speak for itself in the tables and provided our specific knowledge and hypotheses in the discussion addressing relevant differences and trends.

(Page 5, line 138) Although the authors mentioned that “CAD had the lowest prevalence and accounted for less than 2 % (24,242); co-prevalence increased, however, up to 9 % (161,658) 139 in 3-vessel coronary artery disease in the same patient group.”. However, these results were not found in the figures or tables. The authors should show them in the figures or tables.

Response to R2.3:
We like to thank Reviewer 2 for this relevant comment. During the preparation of this manuscript we were confronted with the large pool of data including not only the relevant differences between PAD patients with and without concomitant CAD but also their changes over time. Each of the tables in the current version of the manuscript could be extended 10 times reflecting the specific changes between 2009 and 2018. For example, the co-prevalence (only percentages) looks like this:

    2009 2010 2011 2012 2013 2014 2015 2016 2017 2018
PAD IIb CAD 0 1.5 1.4 1.4 1.5 1.6 1.3 1.4 1.4 1.3 1.2
  CAD 1 2.8 2.7 2.9 3.1 3.1 3.3 3.6 3.5 3.8 3.8
  CAD 2 3.6 3.8 3.7 3.8 3.8 3.8 4.3 4.4 4.5 4.4
  CAD 3 6.8 7.0 7.0 7.3 7.9 8.0 8.6 9.2 9.2 9.1

PAD III CAD 0 1.5 1.5 1.4 1.6 1.5 1.4 1.3 1.5 1.3 1.5
  CAD 1 2.7 2.7 3.2 3.4 3.2 3.5 3.9 4.0 3.8 4.0
  CAD 2 3.8 4.1 3.9 4.0 4.2 4.5 4.6 4.9 5.0 4.8
  CAD 3 7.9 8.1 8.5 8.8 9.2 9.6 9.5 10.8 10.8 10.8

PAD IVu CAD 0 1.6 1.5 1.4 1.4 1.3 1.3 1.2 1.4 1.2 1.1
  CAD 1 2.1 2.1 2.3 2.6 2.9 3.1 3.3 3.7 3.8 4.0
  CAD 2 3.1 3.4 3.3 3.4 3.7 3.9 4.1 4.5 4.4 4.9
  CAD 3 7.1 7.8 7.9 8.3 8.9 9.7 10.4 11.1 11.0 11.5

PAD IVg CAD 0 1.5 1.4 1.3 1.3 1.3 1.1 1.1 1.2 1.2 1.0
  CAD 1 2.2 2.1 2.4 2.7 2.9 3.0 3.5 3.7 3.9 3.9
  CAD 2 3.1 3.4 3.4 3.8 4.0 4.0 4.4 4.7 4.9 4.8
  CAD 3 7.5 8.0 8.8 9.5 10.2 11.4 11.5 12.3 12.5 12.9

Total CAD 0 1.5 1.4 1.3 1.4 1.5 1.3 1.3 1.4 1.3 1.2
  CAD 1 2.5 2.5 2.7 2.9 3.0 3.2 3.6 3.7 3.8 3.9
  CAD 2 3.4 3.7 3.6 3.8 3.9 4.0 4.3 4.5 4.6 4.6
  CAD 3 7.2 7.5 7.8 8.2 8.8 9.2 9.7 10.4 10.4 10.6

And this is not presented separately for males/females. We therefore chose to condense the information into the current table 2 and add it solely in the text to avoid redundance. We aimed to provide a clear and straightforward approach to present the situation of PAD patients with and without CAD. However, if the Reviewers and the Editors think, that this information is too relevant to only provide it in the text, we could offer to add it as a supplementary table.

R2.4:    
The authors need to create a table or figure regarding patient in-hospital mortality and the details of the death, such as cardiac cause.

Response to R2.4:
Thank you for this thoughtful suggestion, which is an excellent idea. Unfortunately, the data we analyzed provided by the Federal Statistical Office has clearly defined parameters that do not allow details on the patient’s death (i.e. cause of death). We agree that this would be a great addition but using this data it is not possible. We are currently working on other data sources that may provide such information in the future (also including death after hospital leave) but its not possible to include this in the current manuscript.

R2.5:
What is the exact definition of CAD in this paper? It should be clearly mentioned in the method section. 

Response to R2.5:
We agree with Reviewer 2 that it was not completely clear, how CAD was defined. We chose the definition by the International Classification of Diseases 10th revision, but our presentation was not clear enough. We revised the respective paragraph in the methods section to specifically show which codes were included and especially how they were analyzed in the manuscript. For example, the most representative and clinically most relevant groups of non-stenosing and 1- to 3-vessel disease were analyzed separately (CAD 0- CAD 3).

R2.6:
(Page 8, line 200) I can’t understand “stationary care”. Please explain the term appropriately. 

Response to R2.6:
Thank you for this comment. We have changed the ‘Germanized’ term “stationary care” to “in-hospital care” to make it clearer.

R2.7:
Please add sufficient explanation about the reason why the costs of hospitalizations with CAD were higher than those without CAD. Was it due to additional intervention procedure for CAD or additional treatment for their comorbidities? The details need to be added.

Response to R2.7:
Thank you for the thoughtful feedback; it is appreciated. Although the data does not allow drawing direct conclusions, there are several potential reasons that need to be thought of. On the one hand there are direct costs of the respective individual procedures, but the additional comorbidities are also a relevant factor. We point out that patients with additional CAD have more overall comorbidities, and it stands to reason that patients with additional CAD have higher intervention rates and other associated treatment costs. Additionally, as just one example, the increasing use of drug eluting stents to treat CAD is certainly one factor for the higher costs. We added a paragraph in the discussion addressing the reimbursement costs, which are most likely to be explained by additional costs e.g. for treating comorbidities.

R2.8:
Since atherosclerosis is a systemic chronic inflammatory vascular disease, it is reasonable that severe PAD patients have greater incidence of CAD as well. Indeed, the conclusion derived from this study seems reasonable but have already shown in the similar studies in the past. What were new findings of the study compared with previous reports? This point needs to be clearly mentioned in the discussion section.

Response to R2.8:
That's an excellent point. Yes, there are similar studies, albeit with a much smaller sample size and to a muss lesser extent focused on comparing PAD patients with and without concomitant CAD. This point was further outlined in more detail in the discussion.

Reviewer 3 Report

Authors analysed 1.8M patients about the trend of in-hospital treatment of patients with PAD with and without CAD. Their data analysis presents findings having clinical value and impact. I have some minor comments which are listed below for improvement:

1. Section 2 M&M: Please provide details on PAD classification, Fontaine I-IV, with citation of relevant references. 

2. Results:  authors are suggested to provide values whether there is any statistically significant difference when comparing findings between two groups, e.g. with and without CAD. For example, page 3,  line 12-122, any significant differences in median reimbursements costs between them? This also applies to other findings or comparisons. 

Figure 3: I find it hard to understand it. blue line refers to minor amputation, while red line refers to major amputation. However, I see one red and one pink lines but there is no description about all of them in the figure legend. what about CAD and no CAD? Please revise it accordingly. 

3. Discussion is fine with good citation of relevant studies. 

Author Response

Reviewer 3:

Comments and Suggestions for Authors

Authors analysed 1.8M patients about the trend of in-hospital treatment of patients with PAD with and without CAD. Their data analysis presents findings having clinical value and impact. I have some minor comments which are listed below for improvement:

R3.1:
Section 2 M&M: Please provide details on PAD classification, Fontaine I-IV, with citation of relevant references. 

Response to R3.1:
Thank you for this comment. In the method section we added that PAD is “atherosclerosis of the arteries of the extremities” and the ICD classification is explained briefly, e.g. I70.22 (2015-2018): pain-free walking distance of 200 meters or less (Fontaine IIb, Rutherford 2-3); Furthermore, PAD classification is explained in detail in reference 14 and 15.

R3.2:
Results:  authors are suggested to provide values whether there is any statistically significant difference when comparing findings between two groups, e.g. with and without CAD. For example, page 3,  line 12-122, any significant differences in median reimbursements costs between them? This also applies to other findings or comparisons. 

Response to R3.2:
Thank you for this comment, we understand your concern.

Due to the vast set of more than 1.8 million patient data, we are certain that any difference between the groups to be compared, no matter how slight, is significant.

R3.3:
Figure 3: I find it hard to understand it. blue line refers to minor amputation, while red line refers to major amputation. However, I see one red and one pink lines but there is no description about all of them in the figure legend. what about CAD and no CAD? Please revise it accordingly. 

Response to R3.3:
Thank you for this comment. We revised figure 3 for better readability.

R3.4:
Discussion is fine with good citation of relevant studies.

Response to R3.4:
Thank you very much for your positive feedback and the appreciation of our work.

Round 2

Reviewer 2 Report

To the authors,

Most of criticisms which I raised in the first round have been clearly and sufficiently addressed by the authors. Now, I have minor comments as follows.

(Question of prior #2) Statistical analysis

Although the access to the web-link (mentioned by the authors in the rebuttal letter e.g. https://link.springer.com/article/10.1007/s00330-021-08285-y) is not permitted to me, I understood what the authors would like to mention. However, without any explanation regarding the fact that “all variables showed statistical significance between CAD+ and CAD- patients due to the large number of cases in database”, the readers will be confused. Please clearly mention about the results of statistical analysis in somewhere of the manuscript.

(Question of prior #3) Prevalence of CAD in PAD patients overtime

Please add the new table which the authors have created as a supplemental material of this paper. It should make more sense.

(Question of prior #8) Discussion regarding the novelty of this paper

Please fix the sentence as follows;

“The vast sample size and clear focus on this issue increase the validity and reliability of your findings and make your study a valuable contribution to the field.”

“The vast sample size and clear focus on this issue increase the validity and reliability of our findings and make our study a valuable contribution to the field.”

Author Response

Reviewer 2:

R2.1:

To the authors,

Most of criticisms which I raised in the first round have been clearly and sufficiently addressed by the authors. Now, I have minor comments as follows.

(Question of prior #2) Statistical analysis

Although the access to the web-link (mentioned by the authors in the rebuttal letter e.g. https://link.springer.com/article/10.1007/s00330-021-08285-y) is not permitted to me, I understood what the authors would like to mention. However, without any explanation regarding the fact that “all variables showed statistical significance between CAD+ and CAD- patients due to the large number of cases in database”, the readers will be confused. Please clearly mention about the results of statistical analysis in somewhere of the manuscript.

Response to R2.1:
Thanks again for your comments! We apologize for forgetting to include our previous statements in the revised manuscript. We have now clearly added this information in the Methods section so that the reader can better understand our thought process and approach.

R2.2:

(Question of prior #3) Prevalence of CAD in PAD patients overtime

Please add the new table which the authors have created as a supplemental material of this paper. It should make more sense.

Response to R2.2:
Thank you for your input. We added Supplementary Table 1 that shows the co-prevalence of PAD and CAD over time with data for each year and combination.

R2.3:

(Question of prior #8) Discussion regarding the novelty of this paper

Please fix the sentence as follows;

“The vast sample size and clear focus on this issue increase the validity and reliability of your findings and make your study a valuable contribution to the field.”

“The vast sample size and clear focus on this issue increase the validity and reliability of our findings and make our study a valuable contribution to the field.”

Response to R2.3:

We thank you very much for this attentive comment. The sentence was corrected accordingly.